# State-level tracking of COVID-19 in the United States

H. Juliette T. Unwin [1,9 ✉], Swapnil Mishra [1,9], Valerie C. Bradley [2,9], Axel Gandy[3,9], Thomas A. Mellan[1], Helen Coupland[1], Jonathan Ish-Horowicz[3], Michaela A. C. Vollmer[1], Charles Whittaker [1], Sarah L. Filippi[3], Xiaoyue Xi[3], Mélodie Monod[3], Oliver Ratmann[3], Michael Hutchinson[2], Fabian Valka [10], Harrison Zhu[3], Iwona Hawryluk [1], Philip Milton[1], Kylie E. C. Ainslie[1], Marc Baguelin[1,4], Adhiratha Boonyasiri [5], Nick F. Brazeau[1], Lorenzo Cattarino [1], Zulma Cucunuba[1], Gina Cuomo-Dannenburg [1], Ilaria Dorigatti [1], Oliver D. Eales[1], Jeffrey W. Eaton [6], Sabine L. van Elsland[1], Richard G. FitzJohn[1], Katy A. M. Gaythorpe[1], William Green[1], Wes Hinsley[1], Benjamin Jeffrey[1], Edward Knock[1], Daniel J. Laydon [1], John Lees [1], Gemma Nedjati-Gilani[1], Pierre Nouvellet[1,7], Lucy Okell [1], Kris V. Parag [1], Igor Siveroni [1], Hayley A. Thompson[1], Patrick Walker[1], Caroline E. Walters[1], Oliver J. Watson [1,8], Lilith K. Whittles[1], Azra C. Ghani[1], Neil M. Ferguson [1], Steven Riley[1], Christl A. Donnelly [1,2], Samir Bhatt [1 ✉] & Seth Flaxman [3 ✉]

As of 1st June 2020, the US Centres for Disease Control and Prevention reported 104,232 confirmed or probable COVID-19-related deaths in the US. This was more than twice the number of deaths reported in the next most severely impacted country. We jointly model the US epidemic at the state-level, using publicly available death data within a Bayesian hierarchical semi-mechanistic framework. For each state, we estimate the number of individuals that have been infected, the number of individuals that are currently infectious and the time-varying reproduction number (the average number of secondary infections caused by an infected person). We use changes in mobility to capture the impact that non-pharmaceutical interventions and other behaviour changes have on the rate of transmission of SARS-CoV-2. We estimate that $R_t$ was only below one in 23 states on 1st June. We also estimate that 3.7% [3.4%–4.0%] of the total population of the US had been infected, with wide variation between states, and approximately 0.01% of the population was infectious. We demonstrate good 3 week model forecasts of deaths with low error and good coverage of our credible intervals.

[1] MRC Centre for Global Infectious Disease Analysis, Abdul Latif Jameel Institute for Disease and Emergency Analytics (J-IDEA), Imperial College, London, UK. [2] Department of Statistics, University of Oxford, Oxford, UK. [3] Department of Mathematics, Imperial College, London, UK. [4] Department of Infectious Disease Epidemiology, London School of Hygiene and Tropical Medicine, London, UK. [5] NIHR Health Protection Research Unit in Healthcare Associated Infections and Antimicrobial Resistance, Imperial College London, London, UK. [6] MRC Centre for Global Infectious Disease Analysis, Imperial College, London, UK. [7] School of Life Sciences, University of Sussex, Brighton, UK. [8] Department of Laboratory Medicine and Pathology, Brown University, Providence, RI, USA. [9] These authors contributed equally: H. Juliette T. Unwin, Swapnil Mishra, Valerie C. Bradley, Axel Gandy. [10] Unaffiliated: Fabian Valka. ✉email: h.unwin@imperial.ac.uk; s.bhatt@imperial.ac.uk; s.flaxman@imperial.ac.uk

The first death caused by COVID-19 in the United States is currently believed to have occurred in Santa Clara County, California on the 6th February[1]. Throughout March 2020, US state governments implemented a variety of non-pharmaceutical interventions (NPIs), such as school closures and stay-at-home orders, to limit the spread of SARS-CoV-2 and ensure the number of severe COVID-19 cases did not exceed the capacity of the health system. In April 2020, the number of deaths attributed to COVID-19 in the United States (US) surpassed that of Italy[2]. Courtemanche et al.[3] used an event-study model to determine that such NPIs were successful in reducing the growth rate of COVID-19 cases across US counties. We similarly seek to estimate the impact of NPIs on COVID-19 transmission, but through a semi-mechanistic Bayesian model that reflects the underlying process of disease transmission and relies on mobility data released by companies such as Google[4].

Mobility measures revealed stark changes in behaviour following the large-scale government interventions in the first stage of the epidemic, with individuals spending more time at home and correspondingly less time at work, at leisure centres, shopping, and on public transit[4,5]. As states continued to ease the stringency of their NPIs in the end of June, policy decisions relied on the interaction between mobility and NPIs and their subsequent impact on transmission, alongside other measures to track and curtail SARS-CoV-2 transmission.

We introduced a new Bayesian statistical framework for estimating the rate of transmission and attack rates for COVID-19 in Flaxman et al.[6]. In that paper, we inferred the time-varying reproduction number, $R_t$, or the average number of people an infected person will infect over time. We calculated the number of new infections through combining previous infections with the generation interval (the distribution of times between infections) and chose the number of deaths to be a function of the number of infections and the infection fatality ratio (IFR). We estimated the posterior probability of our parameters given the observed data, while incorporating prior uncertainty. This made our approach empirically driven, whilst incorporating uncertainty. This approach has also been implemented for Italy[7] and Brazil[8].

In this paper, we extend the Flaxman et al.[6] framework to model transmission in the US at the state-level and include reported cases in our model. We parameterise $R_t$ as a function of several mobility types and include an autoregressive term to capture changes in transmission that are decoupled from mobility, for example hand-washing, social distancing and changes in transmission that are decoupled from mobility. We utilise partial pooling of parameters, where information is shared across all states to leverage as much signal as possible, but individual effects are also included for state and region-specific idiosyncrasies. In this paper, we infer plausible upper and lower bounds (Bayesian credible interval summaries of our posterior distribution) of the total population that had been infected by COVID-19 on 01 June 2020 (also called the cumulative attack rate or attack rate) and estimate the effective number of individuals currently infectious given our generation distribution. We also present effect sizes of the mobility covariates and make short-term forecasts, which we compare with reality throughout June. Details of the data sources and a technical description of our model are found in sections "Methods" and "Data", respectively. General limitations of our approach are presented in the conclusions.

## Results

**Infections**. The percentage of the total population across the US infected by COVID-19 was 3.7% [3.4%–4.0%] on 01 June 2020. However, this low national average masked a stark heterogeneity across the states (Table 1). New York and New Jersey had the highest estimated cumulative attack rates, of 15.9% [12.4%–19.9%] and 14.8% [11.2%–18.2%] respectively, and Connecticut and Massachusetts both had cumulative attack rates over 10%. Conversely, other states that have drawn attention for early outbreaks, such as California, Washington, and Florida, only had cumulative attack rates of around 2% and states that were only in the early stages of their epidemics, like Maine, had estimated cumulative attack rates of <1%.

Figure 1 shows the effective number of infectious individuals and the number of newly infected individuals on any given day up until 01 June 2020 for each of the 8 regions in our model, which are based on US census regions (see Supplementary Note 1 for further descriptions of our groupings). The effective number of infectious individuals is calculated using the generation time distribution, where individuals are weighted by how infectious they are over time, see section "Generated quantities" for more information. The fully infectious average includes asymptomatic and symptomatic individuals. On 01 June 2020, we estimate that there were 41,100 [34,500–46,800] infectious individuals across the US, which corresponds to 0.01% of the population. Table 1 shows estimates of the number of new infections across each states on 01 June 2020. By this date, the estimated number infections were beginning to increase in the Pacific (Alaska, California, Hawaii, Oregon and Washington) and Mountain (Arizona, Colorado, Idaho, Montana, Nevada, New Mexico, Utah and Wyoming) regions.

Our model includes a state-level parameter for the infection ascertainment ratio, IAR, which we define as the number of reported cases divided by the true number of infections (including asymptomatic infections). We only estimate this parameter from 11 May 2020 when more than 375,000 tests are done each day, see Supplementary Note 2 for further information. Column 3 of Table 1 shows the value of the infection ascertainment ratio in our model (see section "Methods") and varies significantly between state. We would not expect the infection ascertainment to be 100% because our model includes asymptomatic individuals who may not know they have COVID-19. The mean value of this ratio varies between 43% (Missouri) to 74% (Kansas and Tennessee), which suggests that states are doing very different levels of testing.

**Reproduction number**. The mean estimate for $R_t$ was below one in 23 states on 01 June 2020 and the 95% credible intervals did not exclude one in any state (see Supplementary Note 3 for $R_t$s by state). Figure 2 depicts the geographical variation in the posterior probability that $R_t$ was <1 using a shape file from the US Census Bureau[9]. The closer a value is to 100%, the more certain we were that the reproduction number was below 1, indicating that new infections were not increasing. The probability was <40% that $R_t < 1$ in 20 states. There was substantial geographical clustering; most states in the Midwest and the South had reproduction numbers that suggested that the epidemic was not under control. We include figures of $R_t$, infections and deaths over time for each state in Supplementary Note 4.

**Model effect sizes**. We find that decreases in the overall average number of visits to different places had a significant effect on reducing transmission. If mobility stopped entirely (100% reduction in average mobility) then $R_t$ would be reduced by 55.1% [26.5%–77.0%]. The country effect size estimates are given in Fig. 3, with regional and state-level effects given in Supplementary Note 5. However, in the US, the average mobility covariate never approached a 100% reduction, and only about half the states had reductions below 50% of the baseline. We define the baseline as the pre-epidemic mobility for each state[4]. As an example,

**Table 1 Posterior model estimates of percentage of total population ever infected, mean new infections per day over week ending 01 June 2020, and infection ascertainment ratio as of 01 June 2020. We present the mean and the 95% credible intervals in square brackets.**

| States | % of total population infected | Estimated mean new infections per day over week ending 01 June 2020 | Infection ascertainment ratio |
|---|---|---|---|
| Alabama | 1.8% [1.4%-2.3%] | 1065 [300-2400] | 59% [35%-80%] |
| Alaska | 0.1% [0.0%-0.2%] | 20 [0-100] | 69% [46%-88%] |
| Arizona | 1.6% [1.2%-2.0%] | 1003 [400-1800] | 55% [35%-81%] |
| Arkansas | 0.7% [0.5%-1.0%] | 451 [100-900] | 66% [45%-86%] |
| California | 1.5% [1.1%-1.9%] | 4863 [2100-10,600] | 59% [37%-80%] |
| Colorado | 3.2% [2.6%-4.1%] | 674 [200-1400] | 54% [33%-79%] |
| Connecticut | 11.4% [9.1%-14.5%] | 520 [200-1200] | 53% [32%-78%] |
| Delaware | 4.4% [3.4%-5.6%] | 153 [0-300] | 68% [45%-87%] |
| District of Columbia | 9.7% [7.6%-12.3%] | 134 [0-300] | 60% [39%-83%] |
| Florida | 1.2% [0.9%-1.5%] | 1350 [600-2700] | 61% [39%-83%] |
| Georgia | 2.7% [2.1%-3.4%] | 1528 [600-3600] | 46% [25%-70%] |
| Hawaii | 0.1% [0.0%-0.3%] | 2 [0-100] | 69% [49%-89%] |
| Idaho | 0.6% [0.4%-0.9%] | 47 [0-100] | 70% [48%-88%] |
| Illinois | 5.2% [4.1%-6.5%] | 2198 [800-4500] | 63% [40%-84%] |
| Indiana | 3.8% [3.1%-4.9%] | 779 [300-1700] | 61% [36%-82%] |
| Iowa | 2.3% [1.7%-2.8%] | 542 [200-1100] | 58% [36%-81%] |
| Kansas | 1.1% [0.8%-1.4%] | 189 [0-400] | 74% [58%-91%] |
| Kentucky | 1.2% [0.9%-1.6%] | 359 [100-800] | 58% [36%-81%] |
| Louisiana | 7.1% [5.7%-9.0%] | 660 [300-1400] | 63% [38%-86%] |
| Maine | 0.7% [0.5%-1.0%] | 78 [0-200] | 64% [42%-85%] |
| Maryland | 5.5% [4.3%-6.7%] | 1675 [600-3200] | 60% [38%-83%] |
| Massachusetts | 11.2% [9.0%-14.0%] | 3387 [1,300-7000] | 43% [23%-68%] |
| Michigan | 5.8% [4.5%-7.2%] | 641 [200-1500] | 54% [30%-76%] |
| Minnesota | 2.6% [1.9%-3.2%] | 1110 [400-2400] | 57% [36%-80%] |
| Mississippi | 3.1% [2.5%-4.1%] | 687 [300-1600] | 48% [27%-73%] |
| Missouri | 1.5% [1.1%-1.9%] | 504 [200-1100] | 43% [24%-69%] |
| Montana | 0.2% [0.0%-0.3%] | 11 [0-100] | 71% [47%-87%] |
| Nebraska | 1.5% [1.2%-2.0%] | 379 [100-900] | 73% [53%-90%] |
| Nevada | 1.8% [1.4%-2.3%] | 197 [0-400] | 62% [40%-84%] |
| New Hampshire | 2.0% [1.5%-2.6%] | 152 [0-400] | 54% [30%-78%] |
| New Jersey | 14.8% [11.2%-18.2%] | 1493 [500-3200] | 52% [31%-79%] |
| New Mexico | 2.0% [1.6%-2.6%] | 176 [0-400] | 61% [36%-81%] |
| New York | 15.9% [12.4%-19.9%] | 2056 [800-4200] | 59% [37%-81%] |
| North Carolina | 1.3% [1.0%-1.7%] | 1859 [800-4100] | 56% [34%-78%] |
| North Dakota | 1.2% [0.8%-1.7%] | 49 [0-200] | 71% [47%-88%] |
| Ohio | 2.1% [1.7%-2.7%] | 1141 [400-2700] | 48% [28%-75%] |
| Oklahoma | 1.1% [0.8%-1.4%] | 117 [0-300] | 66% [43%-85%] |
| Oregon | 0.4% [0.3%-0.6%] | 85 [0-200] | 72% [50%-89%] |
| Pennsylvania | 4.4% [3.4%-5.5%] | 1310 [400-2600] | 51% [28%-78%] |
| Rhode Island | 7.5% [5.8%-9.4%] | 246 [0-700] | 51% [27%-74%] |
| South Carolina | 1.3% [0.9%-1.8%] | 743 [200-1400] | 51% [30%-78%] |
| South Dakota | 1.1% [0.7%-1.5%] | 110 [0-300] | 69% [48%-87%] |
| Tennessee | 0.8% [0.6%-1.1%] | 406 [100-800] | 74% [54%-90%] |
| Texas | 0.9% [0.7%-1.2%] | 2,208 [1000-4400] | 65% [44%-86%] |
| Utah | 0.8% [0.6%-1.1%] | 420 [100-800] | 66% [45%-86%] |
| Vermont | 0.9% [0.5%-1.3%] | 6 [0-100] | 69% [46%-87%] |
| Virginia | 2.3% [1.8%-2.9%] | 1879 [800-3900] | 62% [40%-83%] |
| Washington | 1.8% [1.4%-2.3%] | 533 [200-1200] | 62% [38%-83%] |
| West Virginia | 0.5% [0.3%-0.7%] | 70 [0-200] | 65% [43%-86%] |
| Wisconsin | 1.3% [1.0%-1.7%] | 846 [300-1900] | 61% [37%-80%] |
| Wyoming | 0.3% [0.1%-0.6%] | 15 [0-100] | 65% [39%-85%] |
| National | 3.7% [3.4%-4.0%] | 41,100 [34,500-46,800] | |

consider the largest reduction observed, −62% of the baseline (Minnesota on 12 April 2020). The effect on $R_t$ was a reduction of 37% [16%–56%] from the country level effect.

Increased time spent in residences also reduced transmission; if time spent in residences increased to 100% of the baseline, $R_t$ would be reduced by 15.3% [−27.5% to 54.6%]. Time spent in residences increased by 20% or more from the baseline in 36 states. As an example, consider the largest reduction observed, a 33% increase from the baseline (New Jersey on 10 April 2020).

The effect on $R_t$ from this was a reduction of 5% [−10% to 20%] in New Jersey from the country level effect.

Average mobility and residential mobility are no doubt correlated—when people spend less time in public spaces, captured by our average mobility metric, they conversely spend more time at home. Owing to this collinearity, our model is unable to distinguish between the independent contributions of these covariates, with most of the effect assigned to the average mobility coefficient, due to its greater explanatory power. As a

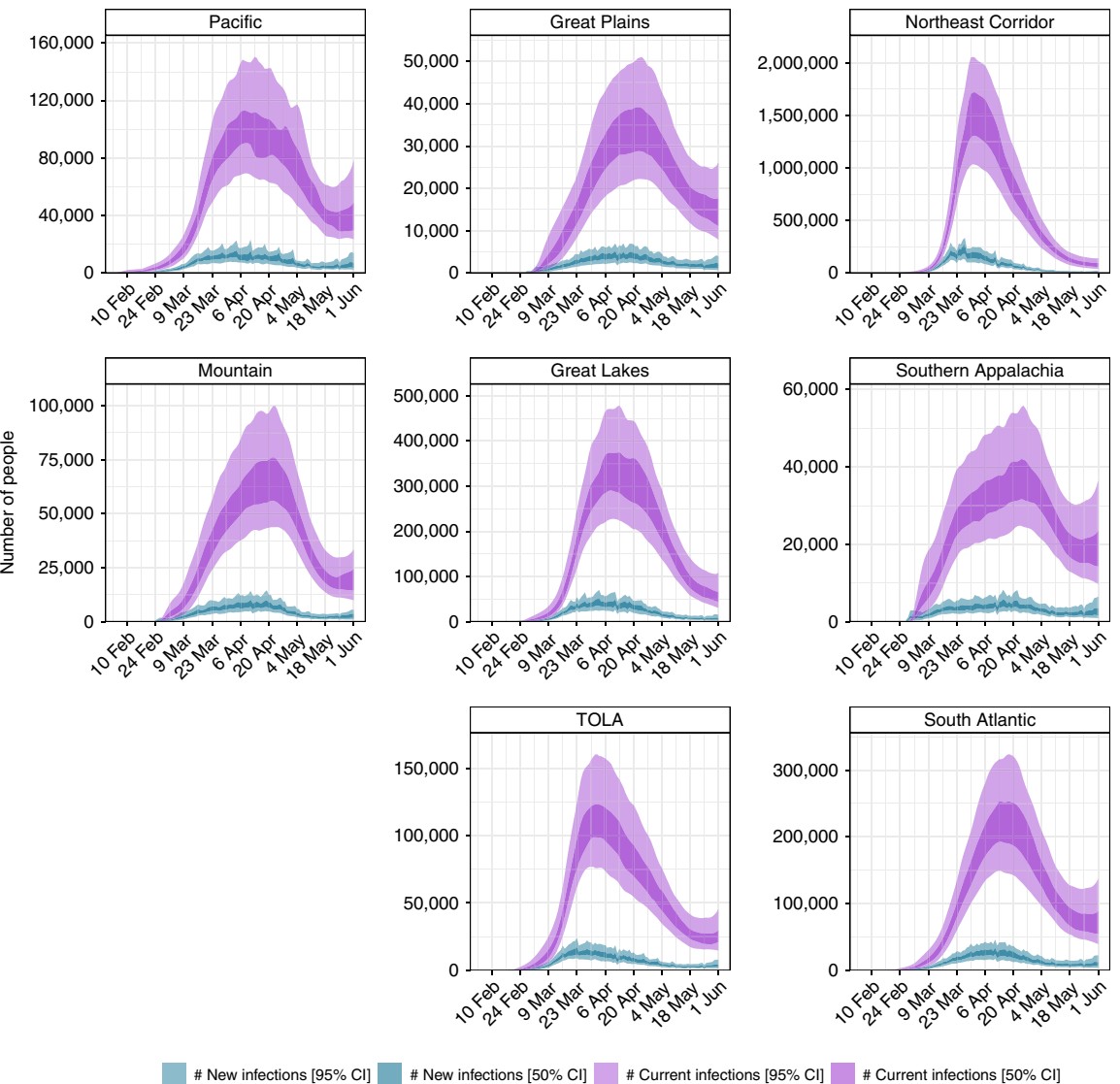

**Fig. 1 Daily estimates of the number of infectious (those still able to transmit) individuals and newly infected individuals.** The light purple band is the 95% credible interval (CI) of the number of infected individuals, dark purple the 50% CI of the number of infected individuals, light blue the 95% CI of the newly infected individuals and dark blue the 50% CI of the newly infected individuals.

check that our overall findings were not biased by this collinearity, we verified that the posterior estimates of these coefficients were not correlated.

**Short-term forecasts**. We used our model to produce short-term death forecasts. Figure 4 compares our forecasts for the 3 weeks after 01 June 2020 (blue line with shaded uncertainty intervals) with the recorded daily number of deaths during this period (coral bars). As expected from our $R_t$ values, deaths were noticeably declining in the Northeastern Corridor, where $R_t > 1$, with particularly low error between our forecasts and reality in New York and Connecticut. In the South, we forecast a flattening or slight increase of deaths, especially in Arkansas, Texas and Florida.

We investigated the numerical accuracy of our forecast using three metrics: mean absolute error, continuous ranked probability score (CRPS) and coverage of credible intervals. We fitted our model to three end points: 1 May, 15 May and 1 June and performed 3-week forecasts from each end point. We compared the metric scores with a log-linear "null" model fit to 31 days of data prior to the three specified end points (see Supplementary

Note 6 for further information). We find our model performs similarly to the null model (1 June) or better (15 May), however, our model fit to 1 May is worse than the null model because we only include cases after 11 May in our models. This suggests that including cases improves the forecasting ability of our model and further justifies our inclusion of them. The coverage of our credible intervals is good for all models, in particular our model and the null model fit to 1 June.

**Model selection and sensitivity**. Mobility data provided a proxy for the behavioural changes that occur in response to non-pharmaceutical interventions. Supplementary Note 7 shows the mobility trends for the 50 states and the District of Columbia up until 01 June 2020 (see section "Data" for a description of the mobility dimensions). The median correlation between the observed average mobility and the timing of the introduction of major NPIs (represented as step functions) was ~86% (see Supplementary Note 8). We make no explicit causal link between NPIs and mobility because this relationship is plausibly causally linked by other factors. The mobility trends data suggests that substantial early outbreak in New York state may have led to

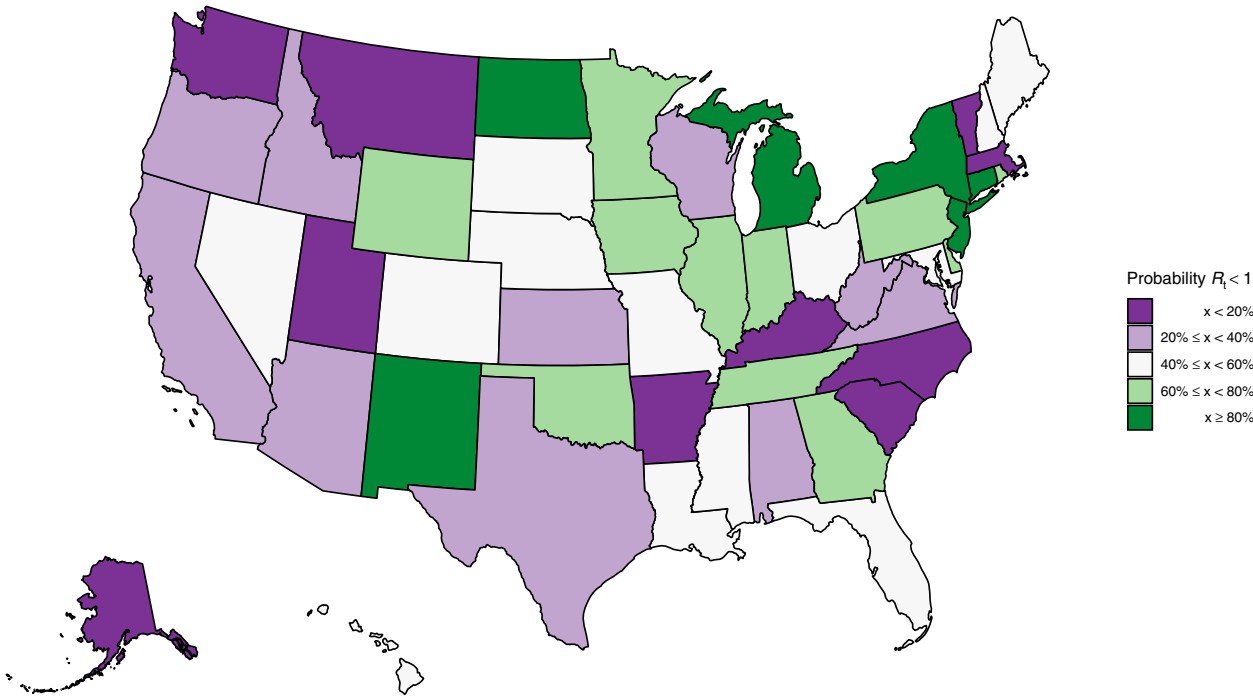

**Fig. 2 Estimates of the probability that the time-varying reproduction number $R_t$ is less than one in each state.** This plot shows the certainty that the rate of transmission is under control. These values are an average over the week ending 01 June 2020.

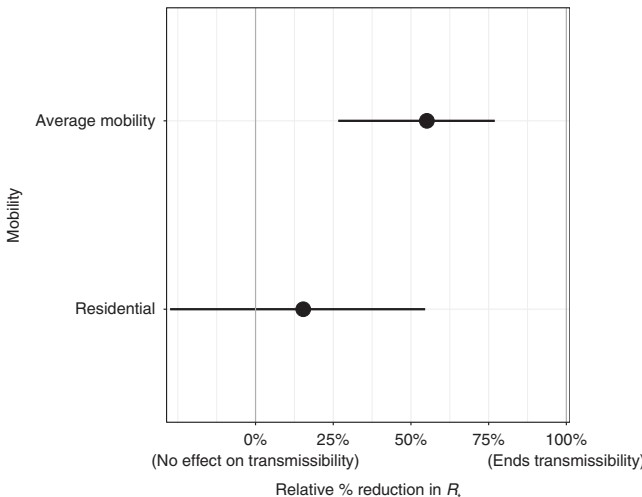

**Fig. 3 Country level covariate effect sizes assuming mobility stopped entirely (100% reduction in average mobility) and residential mobility was increased fully (100% increase in residential mobility).** Average mobility combines "retail & recreation", "grocery & pharmacy", "workplaces". The error bars show 95% credible intervals and the dots show the mean estimate. The sample size $n = 105{,}006$ deaths across the 50 states and the District of Columbia up until 1 June and 479,422 cases from 11 May to 1 June.

substantial changes in mobility in nearby states, like Connecticut, prior to any mandated interventions in those states, which supports including regions in our model. Including both mobility trends and the timing of imposition and lifting of "stay-at-home" orders did not affect the estimated cumulative attack rates (see Supplementary Note 9).

Mobility alone cannot fully capture how transmission evolves over time. In particular, it cannot capture the impact of case-based interventions (such as testing and tracing) or behaviour

changes (such as mask wearing or hand-washing). We use a second-order, weekly, autoregressive process to allow our changes in transmission to be decoupled from mobility. This autoregressive process is an additional term in our parametric equation for $R_t$ and accounts for residual effects by capturing a correlation structure where current $R_t$ is correlated with previous weeks $R_t$. This means that our forecasts were equally good whatever combination of mobility covariates were used because this term could capture the unexplained behaviour. The learnt random effects from this process are shown in Supplementary Note 10 for all states along with the contributions to $R_t$ from the mobility and autoregressive terms for three example states. The autoregressive term increases $R_t$ before lockdown in New York, which could be explained by behaviour such as panic buying. In contrast, the autoregressive term reduces $R_t$ in Montana and could reflect behavioural changes such as hand-washing and self isolation, which can reduce transmission with maintained mobility levels. The autoregressive term remains mostly constant in Washington and suggests that mobility is sufficient to capture the behaviour there.

## Discussion

We developed a Bayesian semi-mechanistic modelling approach to investigate the impact of NPIs on the spread of SARS-CoV-2 in the United States through changes in mobility. Our model relies on death data from the start of the epidemic and recently reported case data to inform our predictions. This enabled us to estimate a realistic infection ascertainment ratio for the 3 weeks before 01 June 2020 for each state, which could help inform policy as to where testing may be lacking. The mean value of this ratio varies between 43% (Missouri) to 74% (Kansas and Tennessee). Our epidemiological grounded mechanistic model links unobserved infections to reported cases and deaths, all within a principled Bayesian statistical framework. This is a significant advancement over curve-fitting models fit directly to reported cases.

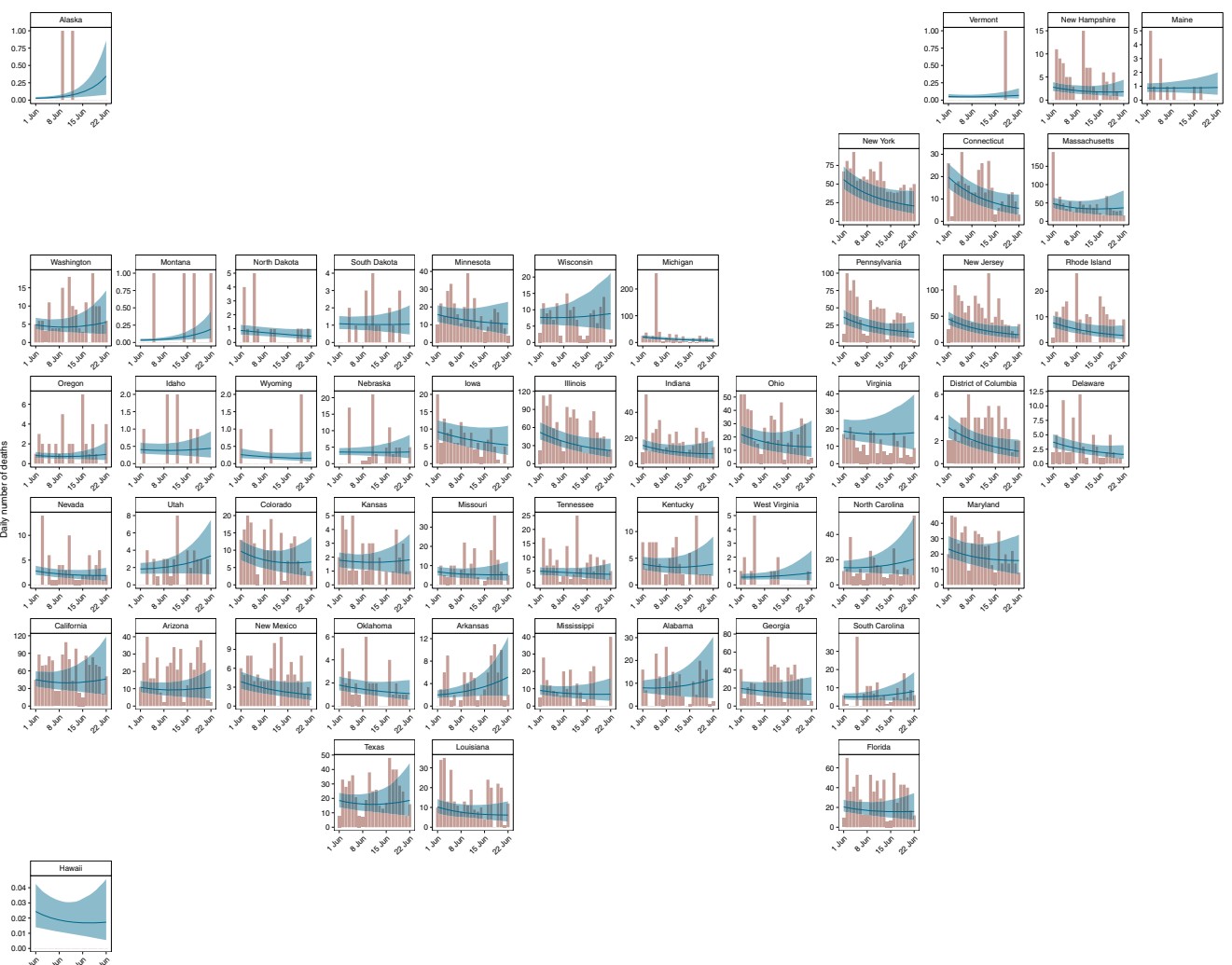

**Fig. 4 Three-week death forecasts for model fitted up until 01 June 2020.** The coral bars show the reported number of deaths for the 3 weeks after 01 June 2020, and the blue line and ribbon show the mean and 95% CI for our forecast estimates.

Our model suggests that although initial reductions in the daily infections had plateaued in most states by 01 June 2020, the reservoir of infectious individuals still remained large with approximately 0.01% of the population being infectious on that date. Despite this, the cumulative attack rate across the US still remained low. We found our attack rate for New York was in line with those from recent serological studies[10]. There is now evidence that mild infection is able to lead to robust immunity (via T cells) but potentially not induce antibody production, which are detected in serosurveys[11]. Therefore, serosurveys might underestimate exposure, particularly in mild cases, and our model may provide an alternative way to measure population exposure. Our cumulative attack rates are, however, sensitive to the assumed values of infection fatality rate (IFR). We account for each individual state's age structure, and further adjust for contact mixing patterns[12], but age-specific modelling may be necessary to capture potential changes in the demographics of cases in states such as Texas, Florida and South Carolina where there is evidence that younger people than were infected at the start of the epidemic are being infected[13,14].

We estimated that 23 states had a posterior mean reproduction number $R_t$ below one on 01 June 2020 and in no states were we more than 95% confident that $R_t$ was below one. We compared our estimates with predictions made by rt.live[15] who use a method that fits the most likely $R_t$ curve to the daily new daily

cases (see Supplementary Note 11). Overall, our estimates were weakly correlated ($\rho = 0.42$) with both of us estimating $R_t > 1$ in 23 states (red points), including Montana and Alaska. However, the rt.live estimates are slightly more pessimistic because they predict $R_t > 1$ in ten states where we predict $R_t < 1$ (blue points). In contrast we predict $R_t > 1$ in five states where they predict $R_t < 1$ (green points). Both sets of reproduction numbers strongly implied that the US epidemic was not under control in many states, and that in the presence of continued migration and the loosening of interventions seen in June, increased infections were to be expected with high probability. We found that state with high reproduction numbers on 01 June 2020 were geographically clustered in the west and south US, whilst the states that had suffered high COVID-19 mortality (such as the Northeast Corridor) in the early phase of the epidemic had lower reproduction numbers. After the period covered by this study, reported cases began to increase in the US, and seven states (Arizona, Arkansas, California, North Carolina, South Carolina, Tennessee and Texas) had recorded higher levels of hospitalisations in early July than before[16,17]. This suggests our estimates that $R_t$ was not less than one were accurate. More recent estimates of $R_t$, the number of infections, and the number of people currently infectious are presented on our website https://mrc-ide.github.io/covid19usa/.

Our 3-week forecasts of daily deaths were highly accurate, confirming the predictive validity of our modelling approach,

despite our having kept mobility constant during our forecasts. These forecasts, alongside our $R_t$ values, show that the epidemic was not under control at the end of May. The accuracy of our forecasts varied during the epidemic and could be due to our assumption that mobility is kept constant over these 3 weeks. Our forecast would perform worse in weeks where mobility was significantly different to the last week of our model fit. When we include cases in our model, we are able to get similar results to a simple "null" model whilst also being about to estimate effect sizes of different mobility trends. We also compared our cumulative death forecasts with those presented by Friedman et al.[18]. Friedman et al. compared the median absolute percentage error (MAPE) for SEIR and dynamic growth rate types of models for models fit to some point in June. Unlike those models, we find the MAPE of our cumulative death forecasts did not increase significantly over time and our 3-week median cumulative death MAPE across all states (9.9%) was similar to the US estimate from Friedman et al. (4.1–8.6), see Supplementary Note 12 for more information.

Our model uses mobility to predict SARS-COV2 transmission. We find that the timings that non-pharmaceutical interventions were implemented was strongly correlated to changes in mobility. This is similar to findings in Abouk and Heydari[5] who find that statewide stay-at-home orders had the strongest causal impact on reducing social interaction and that these orders significantly increase the presence of individuals at home by about six fold (our "residential mobility trend"). This supports our choice of using mobility instead of the timings of NPIs in this study instead of the times of interventions as in Flaxman et al.[6]. We find that magnitude of the reductions in average mobility, and the resulting increases in residential mobility, are important in determining the size of reduction in $R_t$. This agrees with Wang et al.[19] who use a stochastic age- and risk-structured susceptible-exposed-asymptomatic-symptomatic-hospitalised-recovered (SEAYHR) model to considered the effect of various levels of social distancing. They found that social distancing measures, which reduced non-household contacts by <50%, would not prevent a healthcare crisis and that only their 75% and 90% contact reduction scenarios were projected to enable metropolitan areas to remain within healthcare levels.

While mobility, or social distancing measures, will explain a large amount of the trend in $R_t$, there is likely to be substantial residual variation from other behavioural changes such as mask wearing and hand-washing. We accounted for this residual variation through a second-order, weekly, autoregressive process. This stochastic process captures changes in $R_t$ reflected in the data, but is unable to attribute these changes to other determinants of transmission or interventions. We pool parameters in our model to leverage as much signal in our data as possible and to reflect the conjoined nature of some states, in particular in the Northeastern Corridor. While this sharing can potentially lead to over or under estimation of effect sizes, it also means that a consistent signal for all states can be estimated before that signal is presented in an individual state with little data, such as Alaska and Hawaii. Pooling also increases the robustness of our models to under reporting and time lags[6–8].

## Methods

Flaxman et al.[6] introduced a Bayesian model for estimating the transmission intensity and attack rate (percentage of the population that has been infected) from COVID-19 from the reported number of deaths. This framework used the time-varying reproduction number $R_t$ to inform a latent function for infections, and then these infections, together with probabilistic lags, were calibrated against observed deaths. Observed deaths, while still susceptible to under reporting and delays, comprise a more consistent time series than the reported number of confirmed cases, which are susceptible to changes in the probability of ascertainment over the course of the epidemic as testing strategies changed. Our model code is

available on GitHub. Analysis was done using RStan[20] version 2.19.3 within R version 3.6.3.

We adapted the original Bayesian semi-mechanistic model of the infection cycle to all the states in the US and the District of Columbia to infer the reproduction number over time ($R_t$), plausible upper and lower bounds (95% Bayesian credible intervals) of the total populations infected (attack rates) and the number of people currently infected on 01 June 2020. In this paper, we also include the reported number of cases after 11 May 2020, see Supplementary Note 13. This reflects the point in time when over 375,000 tests were being done each day across the US. We include this in our likelihood but do not use them to calculate transmission directly. We parametrise $R_t$ as a function of Google mobility data and include an autoregressive term to capture non-mobility driven behaviour. We fit our model jointly to COVID-19 data from all states to assess the attack rates and number of people who were currently infected. Finally, we use our model to forecast for 3 weeks from 01 June 2020 and compare our estimates of deaths to those recorded in the US for each state. We assume mobility remains constant at the previous value of mobility on the same day the previous week in our forecasts and the autoregressive term remains constant.

**Data**. Our model uses daily real-time state-level aggregated data published by New York Times (NYT)[21] for New York State and John Hopkins University (JHU)[2] for the remaining states. We include 105,006 deaths in our model up until 1 June and 479,422 cases from 11 May to 1 June. Age-specific population counts were drawn from the U.S. Census Bureau in 2018[22] to estimate state-specific infection fatality ratio reflective of the population age structure. The timing of NPIs were collated by the University of Washington[23]. We used Google's COVID-19 Community Mobility Report[4], which provides data on movement in the US by states and highlights the percent change in visits to:

- Grocery & pharmacy: mobility trends for places like grocery markets, food warehouses, farmers markets, speciality food shops, drug stores, and pharmacies.
- Parks: mobility trends for places like local parks, national parks, public beaches, marinas, dog parks, plazas, and public gardens.
- Transit stations: mobility trends for places like public transport hubs such as subway, bus, and train stations.
- Retail & recreation: mobility trends for places like restaurants, cafes, shopping centres, theme parks, museums, libraries, and movie theatres.
- Residential: mobility trends for places of residence.
- Workplaces: mobility trends for places of work.

The residential data includes length of stay at different places compared to a baseline, whereas the other mobility trends are based on number of visits to a certain place. These trends are, therefore, relative, i.e., mobility of −20% means that, compared to normal circumstances individuals are engaging in a given activity 20% less.

**Model specifics**. The true number of infected individuals, $i$, is modelled using a discrete renewal process. We specify a generation distribution $g$ with density $g(\tau)$ as:

$$g \sim \text{Gamma}(6.5, 0.62). \tag{1}$$

Given the generation distribution, the number of infections $i_{t,m}$ on a given day $t$, and state $m$, is given by the following discrete convolution function:

$$
\begin{aligned}
i_{t,m} &= S_{t,m} R_{t,m} \sum_{\tau=0}^{t-1} i_{\tau,m} g_{t-\tau}, \\
S_{t,m} &= 1 - \frac{\sum_{j=0}^{t-1} i_{j,m}}{N_m},
\end{aligned}
\tag{2}
$$

where the generation distribution is discretised by $g_s = \int_{s-0.5}^{s+0.5} g(\tau)d\tau$ for $s = 2, 3, \ldots$, and $g_1 = \int_0^{1.5} g(\tau)d\tau$. The population of state $m$ is denoted by $N_m$. We include the adjustment factor $S_{t,m}$ to account for the number of susceptible individuals left in the population.

Both deaths and cases are observed in our model. We define daily deaths, $D_{t,m}$, for days $t \in \{1, \ldots, n\}$ and states $m \in \{1, \ldots, M\}$. These daily deaths are modelled using a positive real-valued function $d_{t,m} = \mathbb{E}[D_{t,m}]$ that represents the expected number of deaths attributed to COVID-19. The daily deaths $D_{t,m}$ are assumed to follow a negative binomial distribution with mean $d_{t,m}$ and variance $d_{t,m} + \frac{d_{t,m}^2}{\psi_1}$, where $\psi_1$ follows a positive half normal distribution, i.e.,

$$D_{t,m} \sim \text{Negative binomial}\left(d_{t,m}, d_{t,m} + \frac{d_{t,m}^2}{\psi_1}\right), \quad t = 1, \ldots, n \tag{3}$$

$$\psi_1 \sim \mathcal{N}^+(0, 5). \tag{4}$$

Here, $\mathcal{N}(\mu, \sigma)$ denotes a normal distribution with mean $\mu$ and standard deviation $\sigma$. We say that $X$ follows a positive half normal distribution $\mathcal{N}^+(0, \sigma)$ if $X \sim |Y|$, where $Y \sim \mathcal{N}(0, \sigma)$.

We link our observed deaths mechanistically to transmission as in Flaxman et al.[6]. We use a previously estimated COVID-19 infection fatality ratio (IFR,

probability of death given infection) together with a distribution of times from infection to death $\pi$. Details of this calculation can be found in[24,25]. From the above, every region has a specific mean infection fatality ratio $ifr_m$ (see Supplementary Note 13). To incorporate the uncertainty inherent in this estimate we allow the $ifr_m$ for every state to have additional noise around the mean. Specifically we assume

$$ifr_m^* \sim ifr_m \cdot N(1, 0.1). \tag{5}$$

We believe a large-scale contact survey similar to polymod[12] has not been collated for the USA, so we assume the contact patterns are similar to those in the UK. We conducted a sensitivity analysis, shown in Supplementary Note 13, and found that the IFR calculated using the contact matrices of other European countries lay within the posterior of $ifr_m^*$.

Using estimated epidemiological information from previous studies, we assume the distribution of times from infection to death $\pi$ (infection-to-death) to be the convolution of an infection-to-onset distribution ($\pi'$)[25] and an onset-to-death distribution[24]:

$$\pi \sim \text{Gamma}(5.1, 0.86) + \text{Gamma}(17.8, 0.45). \tag{6}$$

The expected number of deaths $d_{t,m}$, on a given day $t$, for state $m$ is given by the following discrete sum:

$$d_{t,m} = ifr_m^* \sum_{\tau=0}^{t-1} i_{\tau,m} \pi_{t-\tau}, \tag{7}$$

where $i_{\tau,m}$ is the number of new infections on day $\tau$ in state $m$ and where, similar to the generation distribution, $\pi$ is discretized via $\pi_s = \int_{s-0.5}^{s+0.5} \pi(\tau) d\tau$ for $s = 2, 3, \ldots$, and $\pi_1 = \int_0^{1.5} \pi(\tau) d\tau$, where $\pi(\tau)$ is the density of $\pi$.

For every state $m$, we also observe daily cases $C_{t,m}$ after $t_c = 11$ May 2020. Similar to daily deaths, daily cases are modelled using a positive real-valued function $\bar{c}_{t,m} = \mathbb{E}[C_{t,m}]$ that represents the expected number of symptomatic and asymptomatic cases. Again, the daily cases $C_{t,m}$ are assumed to follow a negative binomial distribution but with mean $c_{t,m}$ and variance $c_{t,m} + \frac{c_{t,m}^2}{\psi_2}$, where $\psi_2$ follows a positive half normal distribution, i.e.,

$$C_{t,m} \sim \text{Negative binomial}\left(c_{t,m}, c_{t,m} + \frac{c_{t,m}^2}{\psi_2}\right), \quad t = t_c, \ldots, n, \tag{8}$$

$$\psi_2 \sim \mathcal{N}^+(0, 5). \tag{9}$$

As before, we assume the distribution of times from infection to becoming a case $\pi'$ (infection-to-onset) to be

$$\pi' \sim \text{Gamma}(5.1, 0.86). \tag{10}$$

We add in a new link between our observed daily cases and our estimated daily infections. We use our model to estimate an infection ascertainment ratio ($iar_m$) for each state $m$, which is defined as the number of reported cases divided by the true number of infections (including both symtomatic and asymptomatic infections). This follows a Beta distribution, specifically $u_m \sim \text{Beta}(12, 5)$.

The expected number of cases $c_{t,m}$, on a given day $t$, for state $m$ is given by the following discrete sum:

$$c_{t,m} = iar_m \sum_{\tau=0}^{t-1} i_{\tau,m} \pi'_{t-\tau}, \tag{11}$$

where, again, $c_{\tau,m}$ is the number of new infections on day $\tau$ in state $m$ and where $\pi'$ is discretized via $\pi'_s = \int_{s-0.5}^{s+0.5} \pi'(\tau) d\tau$ for $s = 2, 3, \ldots$, and $\pi'_1 = \int_0^{1.5} \pi'(\tau) d\tau$, where $\pi'(\tau)$ is the density of $\pi'$.

We parametrise $R_{t,m}$ as a linear function of the relative change in time spent and number of visits (from a baseline)

$$R_{t,m} = R_{0,m} \cdot f\left(-\left(\sum_{k=1}^2 X_{t,m,k} \alpha_k\right) - \sum_{l=1}^2 Y_{t,m,l} \alpha_{r(m),l}^{\text{region}} - Z_{t,m} \alpha_m^{\text{state}} - \epsilon_{m,w_m(t)}\right), \tag{12}$$

where $f(x) = 2\exp(x)/(1 + \exp(x))$ is twice the inverse logit function. $X_{t,m,k}$ are covariates that have the same effect for all states, $Y_{t,m,l}$ is a covariate that has region-specific effects, $r(m) \in \{1, \ldots, R\}$ is the region a state is in (see Supplementary Note 7), $Z_{t,m}$ is a covariate that has a state-specific effect and $\epsilon_{m,w_m(t)}$ is a weekly AR (2) process, centred around 0, that captures variation between states that is not explained by the covariates.

The prior distribution for $R_{0,m}$[26] was chosen to be

$$R_{0,m} \sim \mathcal{N}(3.28, \kappa) \text{ with } \kappa \sim \mathcal{N}^+(0, 0.5), \tag{13}$$

where $\kappa$ is the same among all states.

In the analysis of this paper we chose the following covariates: $X_{t,m,1} = M_{t,m}^{\text{average}}$, $X_{t,m,2} = M_{t,m}^{\text{residential}}$, $Y_{t,m,1} = 1$ (an intercept), $Y_{t,m,2} = M_{t,m}^{\text{average}}$ and $Z_{t,m} = M_{t,m}^{\text{average}}$, where the mobility variables are from[4] and defined as follows (all are encoded so that 0 is the baseline and 1 is a full reduction of the mobility in this dimension):

- $M_{t,m}^{\text{average}}$ is an average of retail and recreation, groceries and pharmacies, and workplaces. An average is taken as these dimensions are strongly collinear.
- $M_{t,m}^{\text{residential}}$ are the mobility trends for places of residences.

We include regional, as well as state-level parameters, in our model to encapsulate the connected nature of states. This was particularly important in the Northeasten corridor where residents in New Jersey and Connecticut regularly commuted into New York, the early epicentre of the US epidemic (see Supplementary Note 1 for a map of the regions). Regions are based on US Census Divisions, modified to account for coordination between groups of state governments[27].

We assume that seeding of new infections begins 30 days before the day after a state has cumulatively observed 10 deaths. From this date, we seed our model with 6 sequential days of an equal number of infections: $i_{1,m} = \ldots = i_{6,m} \sim \text{Exponential}\left(\frac{1}{\tau}\right)$, where $\tau \sim \text{Exponential}(0.03)$. These seed infections are inferred in our Bayesian posterior distribution.

The weekly, state-specific effect is modelled as a weekly AR(2) process, centred around 0 with stationary standard deviation $\sigma_w$ that, in every state, starts on the first day of its seeding of infections, i.e., 30 days before a total of 10 cumulative deaths have been observed in this state. The AR(2) process starts with $\epsilon_{1,m} \sim \mathcal{N}(0, \sigma_w^*)$,

$$\epsilon_{w,m} \sim \mathcal{N}(\rho_1 \epsilon_{w-1,m} + \rho_2 \epsilon_{w-2,m}, \sigma_w^*) \text{ for } m = 2, 3, 4, \ldots \tag{14}$$

with independent priors on $\rho_1$ and $\rho_2$ that are normal distributions conditioned to be in $[0, 1]$; the prior for $\rho_1$ is a $\mathcal{N}(0.8, 0.05)$ distribution conditioned to be in $[0, 1]$ and the prior for $\rho_2$ is a $\mathcal{N}(0.1, 0.05)$ distribution, conditioned to be in $[0, 1]$. The prior for $\sigma_w$, the standard deviation of the stationary distribution of $\epsilon_w$ is chosen as $\sigma_w \sim \mathcal{N}^+(0, 0.2)$. The standard deviation of the weekly updates to achieve this standard deviation of the stationary distribution is $\sigma_w^* = \sigma_w \sqrt{1 - \rho_1^2 - \rho_2^2 - 2\rho_1^2 \rho_2/(1 - \rho_2)}$. The conversion from days to weeks is encoded in $w_m(t)$. Every 7 days, $w_m$ is incremented, i.e., we set $w_m(t) = \lfloor (t - t_m^{\text{start}})/7 \rfloor + 1$, where $t_m^{\text{start}}$ is the first day of seeding. We keep the AR (2) process constant over the last 7 days of observations since this is less informed by data due to the lags and also over the forecast period.

The prior distribution for the shared coefficients were chosen to be

$$\alpha_k \sim \mathcal{N}(0, 0.5), k = 1, \ldots, 3, \tag{15}$$

and the prior distribution for the pooled coefficients were chosen to be

$$\alpha_{r,l}^{\text{region}} \sim \mathcal{N}(0, \gamma_r), r = 1, \ldots, R, l = 1, 2, \text{ with } \gamma_r \sim \mathcal{N}^+(0, 0.5), \tag{16}$$

$$\alpha_m^{\text{state}} \sim \mathcal{N}(0, \gamma_s), m = 1, \ldots, M \text{ with } \gamma_s \sim \mathcal{N}^+(0, 0.5). \tag{17}$$

We estimated parameters jointly for all states in a single hierarchical model. Fitting was done in the probabilistic programming language Stan[20] using an adaptive Hamiltonian Monte Carlo (HMC) sampler.

**Generated quantities.** The effective number of infectious individuals, $i^*$, on a given day considers how infectious a previously infected individual is on a given day and includes both asymptotic and symptomatic individuals. It is calculated by first re-scaling the generation distribution by its maximum, i.e., $g_\tau^* = \frac{g_\tau}{\max_t g_t}$. Based on (2), the number of infectious individuals is then calculated from the number of previously infected individuals, $c$, using the following:

$$i_{t,m}^* = \sum_{\tau=0}^{t-1} i_{\tau,m} g_{t-\tau}^*, \tag{18}$$

where $i_{t,m}$ is the number of new infections on day $t$ in state $m$.

**Reporting summary.** Further information on research design is available in the Nature Research Reporting Summary linked to this article.

## Data availability
All data necessary for the replication of our results is collated in https://github.com/ImperialCollegeLondon/covid19model. The death data originated from John Hopkins University https://github.com/CSSEGISandData/COVID-19 and the New York Times https://github.com/nytimes/covid-19-data.

## Code availability
All code necessary for the replication of our results is collated in https://github.com/ImperialCollegeLondon/covid19model release 10.

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

## Acknowledgements

We would like to thank Amazon AWS and Microsoft AI for health for computational credits and we would like to thank the Stan development team for their ongoing assistance. We would also like to thank David Joerg and Jacob Steinhardt for their comments through Open Review. This research was partly funded by the Imperial College COVID-19 Research Fund and was supported by Centre funding from the UK Medical Research Council under a concordat with the UK Department for International Development, the NIHR Health Protection Research Unit in Modelling Methodology and Community Jameel. H.J.T.U. is funded by Imperial College London through an Imperial College Research Fellowship grant. S.B. acknowledges the NIHR BRC Imperial College NHS Trust Infection and COVID themes, the Academy of Medical Sciences Springboard award and the Bill and Melinda Gates Foundation.

## Author contributions

H.J.T.U., S.M., V.C.B., A.G., S.B. and S.F. conceived and designed the study. H.J.T.U., S.M., T.A.M., M.A.C.V., H.Z. and P.M. performed mobility analysis. J.I.-H., S.L.F. and X.X. contributed to statistical analysis and M.H. and I.H. did other analysis. H.J.T.U., S.M., A.G., T.A.M., H.C., M.A.C.V., V.W., M.M., O.R., S.B. and S.F. contributed to code development. H.J.T.U., M.A.C.V. and S.F. did the plotting. S.M. and F.V. created the website. H.J.T.U., V.C.B., S.B. and S.F. wrote the first draft of the paper. All authors (H.J.T.U., S.M., V.C.B., A.G., T.A.M., H.C., J.I.-H., M.A.C.V., C.W., S.L.F., X.X., M.M., O.R., M.H., F.V., H.Z., I.H., P.M., K.E.C.A., M.B., A.B., N.F.B., L.C., Z.C., G.C.-D., I.D., O.D.E., J.W.E., S.L.E., R.G.F., K.A.M.G., W.G., W.H., B.J., E.K., D.J.L., J.L., G.N.-G., P.N., L.O., K.V.P., I.S., H.A.T., P.W., C.E.W., O.J.W., L.K.W., A.C.G., N.M.F., S.R., C.A.D., S.B. and S.F.) discussed the results and contributed to the revision of the final manuscript.

## Competing interests

The authors declare no competing interests.
