## [Peer Review File · Nature Communications]

REVIEWERS' COMMENTS

Reviewer #1 (Remarks to the Author):

I'm happy with how the authors have responded to my previous comments. This has improved the papers readability and the presentation of the key results.

Reviewer #2 (Remarks to the Author):

I am satisfied that the authors have engaged with my comments and have made thoughtful and positive changes to the manuscript. Now, as well as achieving its own aims, the manuscript provides some extra information about the forecasting power of the semi-mechanistic model. I agree with the authors that some loss in raw forecasting power when moving to a semi-mechanistic model from a pure forecasting model is acceptable if the semi-mechanistic model provides useful parameter estimates.

I am happy to suggest that the manuscript be published and I wish the authors the best of luck in the rest of the publishing process.

All the best,
Joel Hellewell

Reply to reviewers

Reviewer #1 (Remarks to the Author):

State-level tracking of COVID-19 in the United States
Juliette Unwin et al.

General Comments:

Goals were to estimate total numbers infected, those currently infectious, and the evolving reproduction number, for all US states. From this to understand the effectiveness of the impact of NPIs, and to predict the time-course of the epidemic in each state via calculation of the reproduction number. This latter result is perhaps the key finding: that approximately half the states had a reproduction number below 1 at the end of June, but that the other half were in difficulty. It is becoming increasingly important to better understand the dynamics of COVID-19 numbers at a state, country or even city level of detail.

We too believe that it is important to understand COVID-19 dynamics across the USA at the state level is important. We have chosen to present our model at the state level because we believe that it gives a good overview of the progression of the epidemic across the whole of the USA. Our model is available open source and can be easily extended by individuals wishing to investigate specific counties and cities in detail. This has already been done by a group in Tennessee in the USA and others across Europe.

So called second waves have appeared in July, after the paper was submitted. Florida, Catalonia and Melbourne are just some examples. It is unclear from the paper how the Bayesian methods adopted can provide guidance to public health authorities in these settings, and alternative modelling approaches may be needed.

We think our model is suitable to model the second wave. Since we use mobility as a proxy for non-pharmaceutical interventions, we are able to capture the changes in mobility from the second wave. We also include a second order autoregressive term, which captures non-mobility related changes such as the introduction of wearing masks. We include more recent model runs on our website (<https://mrc-ide.github.io/covid19usa/#/>) that are useful to public health authorities because they show the state of the current epidemic in terms of the time varying reproductive number, cumulative numbers of infections and total number currently infected. We decided to present our results up until June 1st because this was just at the turning point of the epidemic when most states were beginning to open up and towards the end of the first wave.

There are, of course, other ways to evaluate the effectiveness of social distancing interventions. These include the use of agent-based models to analyse how effective a range of measures may be in reducing virus transmission, and the significance of their "strength" and timing of activation. There was no discussion as to how the results obtained from the Bayesian statistical approach adopted was better or worse than use of other modelling approaches.

Thank you for your suggestion here. We have added into the discussion and appendices comparison with other types of models. We compared our R_t predictions with the [rt.live](https://www.rtc.live/) website and found that our estimates on 1st June were slightly more optimistic than theirs. We also compared our three week forecasts with SEIR type and dynamic growth models presented in Friedman et al and found comparable cumulative death median absolute percentage error, the metric that was presented in their paper, with our 1 June estimates (Appendix G). In addition we compare our results to a simple "null" log-linear model and find similar performance when we include cases in our model (Appendix F). We present different timings for our forecasts than in the previous submission to show our model error for more distinct phases of the outbreak.

It is clear that a significant amount of effort went into conducting the analyses and deriving the results, and the key researchers involved must be commended for this.

We thank you for appreciating our academic endeavour with this piece of research.

The paper could be significantly improved as follows:

1. By making the results "stand out" better;

Thanks for this suggestion - we have focused on making our key results stand out in the abstract and discussion

2. Contrast the results with some previously published COVID-19 social distancing results generated using alternative prediction methods;

In addition to comparing our forecasts and R_t predictions, we have compared our results in the discussion to other social distancing literature. Abouk and Heydari found that statewide stay-at-home orders had the strongest causal impact on reducing social interaction and that these orders significantly increased the presence of individuals at home by about six fold (our "residential mobility trend"). This supports our choice of using mobility instead of the timings of NPIs in this study instead of the times of interventions as in Flaxman et al.. Wang et al. used a stochastic age- and risk-structured susceptible-exposed-asymptomatic-symptomatic-hospitalized-recovered (SEAYHR) model to consider the effect of various levels of social distancing. They found that social distancing measures that reduced non-household contacts by <50% would not prevent a healthcare crisis and that only their 75% and 90% contact reduction scenarios were projected to enable metropolitan areas to remain within health care levels. This supports our conclusions that the magnitude of the reduction in mobility is important in determining the reduction in R_t . Other studies used event study regression and a longitudinal pretest–posttest comparison group study to find that implementing social distancing policies reduced COVID-19 case growth rates, with "shelter-in-place" or "stay-at-home" orders being the most significant.

3. Tightening-up the text.

The paper is not terribly well written, there are frequent claims without clear substantiation, and typos too. Given the large number of authors there is really no excuse for this. An example is the description of what Figures 2 and 3 show, which is fairly unclear, not helped by an assumed understanding of USA geographical terms such as TOLA, Midwest etc.

We are sorry that our paper contained typos and other issues with presentation. We have now concentrated on backing up our claims thoroughly and tightening up our terminology.

Methods:

Clearly Bayesian inferencing is a technique that does allow us to establish relationships between mortality data and infections, and understanding the dynamic change in infections is fundamental to better managing SARS-CoV-2 transmission. Mortality data is possibly better than diagnosed case data given large differences in testing between states in the USA, and between countries generally, as the authors state. An interesting exercise might have been to determine whether this is in fact the case, however that was not a goal of the reported study.

In Appendix A we show the ratio of reported cases to our model estimates of infections. Here we see in the early part of the epidemic that very small proportions of infectious were reported as cases and so suggests that cases were unsuitable to be used as the primary mechanism in the model. This is not just due to low initial testing, but also because a large proportion of cases are asymptomatic.

Similarly, hospitalization data may be even better than mortality data given the much larger numbers involved.

We agree that hospitalization data has the potential to be better than mobility data so we have tried to include it in our model. Unfortunately the publicly available data we found mostly gives the total number of people in hospital and not daily new admissions, which are necessary for inclusion in our model. We found it impossible to disentangle these new admissions from daily accounts without more information - even when working closely with one state government.

The use of traffic movement as a proxy for reduction in person-to-person contact achieved by social distancing measures (NPI's) is an attractive approach, though it's a fairly coarse measure. It may not account for the situation in large cities where commuting is heavily weighted to use of public transport.

We agree that capturing public transport is important and we found it an especially important covariate for our models in Europe. In an earlier version of our model, we used the "transit" google mobility trend for the states which had significant transit usage. However, we did not find the covariate was significant when including it in our state level model of the US.

The finding that reducing visits to different places reduces transmission is perhaps self-evident and of limited value to public health authorities who are managing rapidly increasing case numbers and healthcare pressures. The term baseline was used frequently, but was it defined?

We refer to the baseline as the pre-covid19 epidemic mobility levels. We have ensured that we define this clearly in the model effect size section.

As I'm not a statistician I am unable to comment on whether the prediction approach adopted does improve on current infection curve fitting methods, as claimed.

Positives: Determining the time-changing reproduction number is an important goal, and this was achieved. A validation exercise determined that predicted death 3 weeks hence successfully matched what actually transpired, which is an important outcome, though extrapolating results to the USA as a whole is probably not that useful from a public health perspective.

Thank you for your comment here - as well as evaluating our forecasts across the whole of the USA, we have also included figures showing our findings at the state level (Appendix F).

A key goal was to determine that NPIs can reduce the growth rate in cases, and this was achieved. But how we keep rates low as we learn to live with on-going coronavirus transmission is highly nuanced and other prediction techniques may address this better.

In our paper we have not aimed to forecasted the impact of different NPIs. Instead we show the effect size of changes in mobility. This is an interesting question, but our model is not designed to do this and would require substantial new innovation and more importantly data.

Challenges: The area that we are wrestling with at the end of July is how to conduct COVID-19 modelling to better inform public health responses. Many countries have started to ease NPIs (i.e. relaxing social distancing measures) and some have seen significant rebounds in cases, hospitalisations and deaths. The desire to restart economies and get them out of "lockdown" has to be balanced against preventing a rapid growth in new cases, and evidence from July alone has seen cities having to reintroduce strict social distancing measures, such as Barcelona and Melbourne. We need to understand how best to reintroduce such measures, their strength (and thus societal impact) and timing. It's unclear that the methods adopted can give such guidance. This is not a criticism of the quality of the research itself, but rather of the value of the results.

A further issue is that data up to 1st June was used, however much has happened in the past two months, as mentioned above. What we have seen is a much more complex waxing and waning of case numbers as NPIs are introduced and relaxed. This probably requires specific social distancing measures to be activated by public health authorities, such as trading off school reopening against a reduction in workplace attendance or community-wide contact. It is unclear how the methods used can help answer such issues.

We chose to present our results up until 1 June as we see this as the transition between phase one and two of the epidemic. We believe our methods could be used to model the second wave. We have a soon to be published age specific model variant (Monod et al. 2020), using the same underlying Bayesian statistics, which starts to tackle these questions. We also show updated estimates on our website (<https://mrc-ide.github.io/covid19usa/#/>) showing how our methods have been able to estimate the time varying reproduction number up until the end of July.

Reviewer #2 (Remarks to the Author):

This is a complex and principled attempt at teasing apart the impacts of various non-pharmaceutical interventions (NPIs) across US states. Indeed, the model has to be complex to account for the fact that each state introduced different NPIs at different times, in different orders, with different intensities and public adherence, at different stages in their own epidemic. The model does this by inferring the hidden trajectory of infections based on recorded deaths (and later, reported cases) and the delays from infection to onset, and onset to death. From this trajectory of infections the model calculates an estimate of how the reproduction number changes over time. The model then attempts to link these changes in the reproduction number to proxy variables that indirectly measure how well the NPIs are working, in this case google mobility data measuring changes in population movement and adherence to "stay in place" orders.

Thanks for reading our manuscript - we believe this is a fair summary of our research.

Region-level parameters

I am curious about the inclusion of region-level parameters. Firstly, how did you decide upon the regions? Are they a region grouping used by some institution? I couldn't find your grouping from a quick google search.

We based our regions on the US census regions and slightly adapted them to capture agreements between states deciding to coordinate their responses.

Secondly, what is the rationale behind including region-level parameters in the model? State-level parameters make sense because that seems to be the level of the governmental machinery that decides upon and implements the NPIs. It would be good to make more explicit why you have separated the states into these regions and why you think states within a region should have shared changes in the reproduction number due to the intercept and mobility term.

We include regional-level parameters in the model because we wanted to capture the fluid borders across the states for work (especially in the northeastern corridor). Also, Governors on East and West coasts formed pacts to decide when to reopen economies and we wanted to be able to reflect this in our model. It was particularly important to include regional level parameters in our model to get good fits with New York and New Jersey early on and our model was sometimes unstable without them.

Mobility data

You average the mobility data that describes changes in the percentage of the population going to various places because you are worried about collinearity and this makes sense. However, you include both the decreases in people going to places other than their house as well as increases in people staying at their house. Would this not also potentially introduce collinearity between these two variables? They seem strongly correlated in every state in Figure 12. Have you checked for signs of collinearity in the model fit and are satisfied that it is not present?

We were concerned about the collinearity of our model, which is why we only used the average mobility and residential effect sizes in the model instead of using each of the google mobility trends separately. When people spend less time in public spaces, captured by our average mobility metric, they conversely spend more time at home. Thus while collinearity in the covariates is likely, we found there was no correlation between our posterior estimates for our country level average mobility and residential coefficients. We found the effect size of residential and average mobility were important for explaining the trends with data up until April and May, so wanted to present effect sizes for both of them.

Forecasting evaluation

I think it would greatly improve the strength of the results in my eyes to present the forecasting scores next to those of some simpler ("null") models. This would demonstrate the power of using the semi-mechanistic model including mobility covariates over using a "pure" forecasting time series model that makes forecasts based on previous values of the response variable. It would be easy to include the same scores for forecasts made on the same date for other models in Figures 10 and 11. Could include "prediction for tomorrow is the same as observation today/last known observation" and an ARIMA (values of AR, I, MA chosen by AIC) model. This would show that the mechanistic part of the model is bringing in some predictive power beyond what can be inferred from past deaths.

Thanks for your suggestion about a "null" model - we have compared our forecasts with a log-linear null model fitted to the past 31 days of data (see Appendix F). We changed the dates we fit our forecasts to be 1 May, 15 May and 1 June to capture different phases of the epidemic. We found similar (1 June) or slightly better (15 May) performance with our model to the null model when we included cases. We were happy with this because our model also estimates the time varying reproduction number and mobility effect sizes.

Model selection

It's clear from reading the manuscript that there has been a good deal of model selection and refinement (seemingly based on forecasting performance) but we are only presented with the final product. I don't think it's always useful for reviewers to demand to see many copies of the same plots for slightly different iterations of a model but it would be nice to see somewhere in the appendix a table that shows different model iterations considered and their respective forecasting scores in terms of MAE, CRPS etc.

Thanks for your useful suggestion here. We added a section to the model selection results where we explain that our forecasts do not vary significantly when we choose different covariates. This is because the autoregressive term captures the unexplained behaviour.

State weekly effects

I am curious about the effect sizes from the weekly, state-level AR part of the model (Figure 15). In some states (Alaska, Wyoming, Hawaii, North Dakota), it predicts anywhere between a 50% increase and 50% reduction in R_t . Am I correct in understanding that this is where the benefit of pooling comes in?

The benefit of pooling enables us to share behaviour from mobility (i.e. mobility effect sizes between states). We have added in an extra figure in Appendix K to further explain the AR part of the model. Here we show the contributions to R_t from each of the mobility and autoregressive terms for three example states. The autoregressive term increases R_t before lockdown in New York, which could be explained by behaviour such as panic buying. In contrast, the autoregressive term reduces R_t in Montana and could reflect behavioural changes such as hand-washing and self isolation, which can reduce transmission with maintained mobility levels. The autoregressive term remains mostly constant in Washington and suggests that mobility is sufficient to capture the behaviour there.

Since the state-level effect sizes for mobility reduction (Figure 9) have much wider credible intervals than on the regional level (Figure 8) and the national level (Figure 6). I think it might be good to state clearly what I just ran through above (if it is correct) when linking to Appendix D in Section 2.3. It might even be worth moving the national effect sizes (Figure 6) out of the Appendix and into the main text as this is one of your main results.

Thanks for suggestions - we have moved the national effect size into the main body of the text and added in some more explanation.

Minor questions regarding methodology:

- Why use two times the inverse logit function in equation 12?

In our first application of our covid19 model to Europe we used an exponential relationship between R_t and our covariates (<https://arxiv.org/pdf/2004.11342.pdf>). We update this to an inverse logit relationship when we include mobility in the Italy report

(<https://www.imperial.ac.uk/media/imperial-college/medicine/mrc-gida/2020-05-04-COVID19-Report-20.pdf>). The exponential link function led to problems in the MCMC sampler that were resolved with the inverse logit function.

- How did you arrive at the prior distributions for the 6 days of infections that seed the outbreak? ($\tau \sim \text{exponential}(0.03)$ and infections $\sim \text{exponential}(1 / \tau)$)

We use the same priors as Flaxman *et al.* Nature (2020) and further information can be found in "Supplementary Discussion 10. Sensitivity of probabilistic seeding scheme". There is a 2-3 week lag between cases and deaths, so we go back 2-3 weeks (30 days back, 6 days of seeding) and try to infer the number of imported cases. It is possible to investigate these priors with the package, *epidemia* (<https://github.com/ImperialCollegeLondon/epidemia>), which we have since developed for our model.

- Have you thought about trying to infer a weekly reporting effect, maybe on a state level? Seems like a lot of the reported deaths in Figure 3 might have pretty considerable weekend reporting effects (see Pennsylvania, Illinois, Arizona).

We have not considered week / weekend reporting effects in this model, but we understand that reporting delays will impact our results. We believe the weekend effects are due to reporting delays at the weekend, which is similar to Europe, and we do not have information in the USA about when the deaths actually occurred. We have considered

the change in week and weekend effects of contacts in an age specific version of the model, which we hope to release onto a preprint server imminently (Monod et al. 2020). We have also investigated aggregating our data at the weekly level for our UK model and found that the trends were similar to our daily model.

Summary

- I think that this manuscript would be improved by demonstrating more clearly that the mechanistic model with mobility covariates provides additional forecasting power on top of models that only infer trends from previous observed deaths. This would then give credibility to your forecasts, R_t estimates, and infection curves.
- In showing forecasting power, you would also demonstrate how mobility data aids accurate forecasting, linking mobility to transmission. While you are reluctant to link changes in mobility to the implementation of NPIs, your results do show that if NPIs reduce mobility (an argument that is not too difficult for others to make), then they will reduce transmission.
- Explain how you grouped states into regions and why you think that states in these regions should have shared effects of mobility and a region-wide intercept.

Thanks for these comments, we have addressed each point above where they have been introduced and made relevant adjustments to the manuscript.

I think this manuscript makes an important contribution to our understanding of the nature of the COVID-19 outbreak in the US. However, I cannot recommend that it is published until the comments that I have made above are addressed. I have also provided an annotated copy of the manuscript to highlight any grammatical errors I noticed while reading.

Thanks for your annotated copy - we have addressed these suggestions in the text.

Sincerely,

Joel Hellewell

REVIEWER COMMENTS

Reviewer #1 (Remarks to the Author):

State-level tracking of COVID-19 in the United States
Juliette Unwin et al.

General Comments:

Goals were to estimate total numbers infected, those currently infectious, and the evolving reproduction number, for all US states. From this to understand the effectiveness of the impact of NPIs, and to predict the time-course of the epidemic in each state via calculation of the reproduction number. This latter result is perhaps the key finding: that approximately half the states had a reproduction number below 1 at the end of June, but that the other half were in difficulty.

It is becoming increasingly important to better understand the dynamics of COVID-19 numbers at a state, country or even city level of detail. So called second waves have appeared in July, after the paper was submitted. Florida, Catalonia and Melbourne are just some examples. It is unclear from the paper how the Bayesian methods adopted can provide guidance to public health authorities in these settings, and alternative modelling approaches may be needed.

There are, of course, other ways to evaluate the effectiveness of social distancing interventions. These include the use of agent-based models to analyse how effective a range of measures may be in reducing virus transmission, and the significance of their "strength" and timing of activation. There was no discussion as to how the results obtained from the Bayesian statistical approach adopted was better or worse than use of other modelling approaches.

It is clear that a significant amount of effort went into conducting the analyses and deriving the results, and the key researchers involved must be commended for this. The paper could be significantly improved as follows:

1. By making the results "stand out" better;
2. Contrast the results with some previously published COVID-19 social distancing results generated using alternative prediction methods;
3. Tightening-up the text.

The paper is not terribly well written, there are frequent claims without clear substantiation, and typos too. Given the large number of authors there is really no excuse for this.

An example is the description of what Figures 2 and 3 show, which is fairly unclear, not helped by an assumed understanding of USA geographical terms such as TOLA, Midwest etc.

Methods:

Clearly Bayesian inferencing is a technique that does allow us to establish relationships between mortality data and infections, and understanding the dynamic change in infections is fundamental to better managing SARS-CoV-2 transmission. Mortality data is possibly better than diagnosed case data given large differences in testing between states in the USA, and between countries generally, as the authors state. An interesting exercise might have been to determine whether this is in fact the case, however that was not a goal of the reported study. Similarly, hospitalization data may be even better than mortality data given the much larger numbers involved.

The use of traffic movement as a proxy for reduction in person-to-person contact achieved by social distancing measures (NPI's) is an attractive approach, though it's a fairly coarse measure. It may not account for the situation in large cities where commuting is heavily weighted to use of public transport.

The finding that reducing visits to different places reduces transmission is perhaps self-evident and of limited value to public health authorities who are managing rapidly increasing case numbers and healthcare pressures. The term baseline was used frequently, but was it defined?

As I'm not a statistician I am unable to comment on whether the prediction approach adopted does improve on current infection curve fitting methods, as claimed.

Positives: Determining the time-changing reproduction number is an important goal, and this was achieved. A validation exercise determined that predicted death 3 weeks hence successfully matched what actually transpired, which is an important outcome, though extrapolating results to the USA as a whole is probably not that useful from a public health perspective. A key goal was to determine that NPIs can reduce the growth rate in cases, and this was achieved. But how we keep rates low as we learn to live with on-going coronavirus transmission is highly nuanced and other prediction techniques may address this better.

Challenges: The area that we are wrestling with at the end of July is how to conduct COVID-19 modelling to better inform public health responses. Many countries have started to ease NPIs (i.e. relaxing social distancing measures) and some have seen significant rebounds in cases, hospitalisations and deaths. The desire to restart economies and get them out of "lockdown" has to be balanced against preventing a rapid growth in new cases, and evidence from July alone has seen cities having to reintroduce strict social distancing measures, such as Barcelona and Melbourne. We need to understand how best to reintroduce such measures, their strength (and thus societal impact) and timing. It's unclear that the methods adopted can give such guidance. This is not a criticism of the quality of the research itself, but rather of the value of the results.

A further issue is that data up to 1st June was used, however much has happened in the past two months, as mentioned above. What we have seen is a much more complex waxing and waning of case numbers as NPIs are introduced and relaxed. This probably requires specific social distancing measures to be activated by public health authorities, such as trading off school reopening against a reduction in workplace attendance or community-wide contact. It is unclear how the methods used can help answer such issues.

Reviewer #2 (Remarks to the Author):

This is a complex and principled attempt at teasing apart the impacts of various non-pharmaceutical interventions (NPIs) across US states. Indeed, the model has to be complex to account for the fact that each state introduced different NPIs at different times, in different orders, with different intensities and public adherence, at different stages in their own epidemic. The model does this by inferring the hidden trajectory of infections based on recorded deaths (and later, reported cases) and the delays from infection to onset, and onset to death. From this trajectory of infections the model calculates an estimate of how the reproduction number changes over time. The model then attempts to link these changes in the reproduction number to proxy variables that indirectly measure how well the NPIs are working, in this case google mobility data measuring changes in population movement and adherence to "stay in place" orders.

Region-level parameters

I am curious about the inclusion of region-level parameters. Firstly, how did you decide upon the regions? Are they a region grouping used by some institution? I couldn't find your grouping from a quick google search. Secondly, what is the rationale behind including region-level parameters in the model? State-level parameters make sense because that seems to be the level of the governmental machinery that decides upon and implements the NPIs. It would be good to make more explicit why you have separated the states into these regions and why you think states within a region should have shared changes in the reproduction number due to the intercept and mobility term.

Mobility data

You average the mobility data that describes changes in the percentage of the population going to various places because you are worried about collinearity and this makes sense. However, you include both the decreases in people going to places other than their house as well as increases in people staying at their house. Would this not also potentially introduce collinearity between these two variables? They seem strongly correlated in every state in Figure 12. Have you checked for signs of collinearity in the model fit and are satisfied that it is not present?

Forecasting evaluation

I think it would greatly improve the strength of the results in my eyes to present the forecasting scores next to those of some simpler ("null") models. This would demonstrate the power of using the semi-mechanistic model including mobility covariates over using a "pure" forecasting time series model that makes forecasts based on previous values of the response variable. It would be easy to include the same scores for forecasts made on the same date for other models in Figures 10 and 11. Could include "prediction for tomorrow is the same as observation today/last known observation" and an ARIMA (values of AR, I, MA chosen by AIC) model. This would show that the mechanistic part of the model is bringing in some predictive power beyond what can be inferred from past deaths.

Model selection

It's clear from reading the manuscript that there has been a good deal of model selection and refinement (seemingly based on forecasting performance) but we are only presented with the final product. I don't think it's always useful for reviewers to demand to see many copies of the same plots for slightly different iterations of a model but it would be nice to see somewhere in the appendix a table that shows different model iterations considered and their respective forecasting scores in terms of MAE, CRPS etc.

State weekly effects

I am curious about the effect sizes from the weekly, state-level AR part of the model (Figure 15). In some states (Alaska, Wyoming, Hawaii, North Dakota), it predicts anywhere between a 50% increase and 50% reduction in R_t . Am I correct in understanding that this is where the benefit of pooling comes in? Since the state-level effect sizes for mobility reduction (Figure 9) have much wider credible intervals than on the regional level (Figure 8) and the national level (Figure 6). I think it might be good to state clearly what I just ran through above (if it is correct) when linking to Appendix D in Section 2.3. It might even be worth moving the national effect sizes (Figure 6) out of the Appendix and into the main text as this is one of your main results.

Minor questions regarding methodology:

- Why use two times the inverse logit function in equation 12?
- How did you arrive at the prior distributions for the 6 days of infections that seed the outbreak? ($\tau \sim \text{exponential}(0.03)$ and $\text{infections} \sim \text{exponential}(1 / \tau)$)
- Have you thought about trying to infer a weekly reporting effect, maybe on a state level? Seems like a lot of the reported deaths in Figure 3 might have pretty considerable weekend reporting effects (see Pennsylvania, Illinois, Arizona).

Summary

- I think that this manuscript would be improved by demonstrating more clearly that the mechanistic model with mobility covariates provides additional forecasting power on top of models that only infer trends from previous observed deaths. This would then give credibility to your forecasts, R_t estimates, and infection curves.
- In showing forecasting power, you would also demonstrate how mobility data aids accurate forecasting, linking mobility to transmission. While you are reluctant to link changes in mobility to

the implementation of NPIs, your results do show that if NPIs reduce mobility (an argument that is not too difficult for others to make), then they will reduce transmission.

- Explain how you grouped states into regions and why you think that states in these regions should have shared effects of mobility and a region-wide intercept.

I think this manuscript makes an important contribution to our understanding of the nature of the COVID-19 outbreak in the US. However, I cannot recommend that it is published until the comments that I have made above are addressed. I have also provided an annotated copy of the manuscript to highlight any grammatical errors I noticed while reading.

Sincerely,
Joel Hellewell